# Clinicopathological comparison of eccrine poroma and porocarcinoma: Ki-67 index is not a decisive factor

Anna-Stiina Meriläinen[1,2]*, Harri Sihto[2,3], Jorma Isola[4¤a], and Virve Koljonen[2,5]

1 Department of Surgery, The Central Hospital of Tavastia Proper, Hämeenlinna, Finland, 2 University of Helsinki, Helsinki, Finland, 3 Department of Pathology Helsinki University Hospital, Helsinki, Finland, 4 Faculty of Medicine and Health Technology, Tampere University, Tampre, Finland, 5 Department of Plastic Surgery Helsinki University Hospital, Helsinki, Finland

¤a Current address: Jilab Inc., Tampere, Finland
* anna-stiina.merilainen@helsinki.fi

## Abstract

### Background

Eccrine poroma (EP) and porocarcinoma (EPC) arise from the intraepidermal part of the sweat gland. Clinically they resemble each other and cannot be distinguished without histopathological examination. EPC has been described as aggressive; however, the Ki-67 index is scarcely reported. The aim of this study was to compare clinicopathological factors between EP and EPC with special interest in Ki-67 index.

### Methods and Findings

50 EP and 22 EPC samples with clinical data from 48 EP and 21 EPC patients were collected from the Finnish Biobanks. We performed immunohistochemistry using a Ki-67 antibody on a tissue microarray and analysed the Ki-67 index with ImmunoRatio 2.5-program. We analysed 48 EP and 21 EPC samples. EPC patients were older (p = 0.019) and their tumours larger (p = 0.003) but other than these there were no statistically significant differences. Ki-67 ratios were similar (medians: EP 0.6% and EPC 0.5%). The median follow-up time in EP group was 12 (range 1.5–30.6 years) and in EPC group 7 years (range 0.75–20.3 years). The survival of EP patients was better than EPC patients but did not reach statistical significance and, in the Cox multivariate analysis only age had statistically significant effect (HR 1.061, 95% CI 1.026–1.099, p < 0.001). Ki-67 index had no statistically significant effect on survival in EPC group in the Cox univariate analysis (HR 0.746, 95% CI 0.390–1.43, p = 0.378).

### Conclusions

EPC patients were older and their tumours larger. There was no difference in Ki-67 index between EP and EPC groups. In the Cox multivariate analysis only age had a

**Data availability statement:** The data were collected from three Finnish Biobanks and under the Finnish Biobank act (688/2012) are not allowed to be shared in Open Access portals. The data are available at each Biobank with reasonable requests from the Biobanks. Helsinki Biobank: https://helsinginbiopankki.fi/en/front-page.ProjectcodeHBP2018003. Tampere Clinical Biobank: https://www.pirha.fi/en/web/english/for-professionals/finnish-clinical-biobank-tampere . Project code 2018-006. The Biobank of Northern Finland, Borealis: https://oys.fi/biopankki/ . Project code BB_2019_3005.

**Funding:** This study was funded by Helsinki University Central Hospital (HUCH) Competitive Research Fund (EVO) Grant number TYH2020208.

**Competing interests:** The authors have declared that no competing interests exist.

statistically significant effect on survival. According to our findings Ki-67 index might not be a decisive factor in the prognosis of EPC. Further studies to validate our current findings are warranted.

## Introduction

Benign and malignant tumours arising from the intraepidermal part of the sweat gland has been called many names and described in the English literature for almost 100 years [1–3]. Nowadays they are called eccrine poroma (EP) and its malignant counterpart eccrine porocarcinoma (EPC).

The clinical appearance of both EP and EPC are similar and therefore cannot be distinguished without histopathological examination. Both tumours can be coloured red/reddish, brown, violaceous, or skin coloured and the formation can be nodulus, or plaque, and there can also be ulceration on the surface [4–13]. The time from first clinical appearance of EPC to diagnosis varies from 2 months to over 40 years [10] but of note, the median times are from 23 to 67 months [14–16]. The slow growth is also reported in EP with time of first appearances to diagnosis ranging from 1 to 10 years [4,6–8,17,18].

Both EP and EPC can be found anywhere in the body. EP is historically thought to be mostly encountered in the palmoplantar area. However, when looking at the literature it seems that EPs are distributed more evenly; 29–35% in the palmoplantar/acral area, 16–50% in the trunk, 16–27% in the head and neck area and 24% in the extremities [4–6,9]. Most common locations of EPC are the lower limb and the head and neck area [10,14,19–22]. EP affects slightly younger patients than EPC according to the literature, mean/median ages 49–66 years [4–6,9,23] and 65–78 years [19–21,24–27], respectively.

In the recent literature metastases in EPC occurs at presentation in 4.0–8.1% [20,27,28] and during follow up 1.9–12.0% [27,28]. In these studies, the disease related deaths vary from 1.9 to 8.1% [27,28]. Local recurrences of EPC are reported from 4 to 22% [10,14,21,27].

Although EPC is described as an aggressive tumour the information on tumour proliferation index is scarcely reported. One of the most used and well-established immunohistochemical nuclear marker depicting tumour cells' proliferation is Ki-67 which is expressed from G1 to M phase of cell division [29]. It has been shown to have a prognostic factor in breast cancer [30] but in skin cancers the role of Ki-67 is still unclear. In Yerebakan et al. study the Ki-67 index was statistically significantly higher in recurrent basal cell carcinoma (BCC) independent of the tumour subtype [31]. More recently, Mendez-Flores et al. reported no differences between the aggressive and less-aggressive subtype groups of BCC [32]. In cutaneous Merkel cell carcinoma (MCC) the Ki-67 index ≥55% was more frequently encountered in patients with progressive disease or patients alive with/dead of disease than in patients with no more disease burden [33]. But when put into multivariate analysis Ki-67 did not have an independent prognostic effect on the disease-specific survival [33]. Considering the difference of Ki-67 index between benign and malignant forms of skin tumours,

in Kim et al. study the Ki-67 index was significantly higher in acral lentiginous melanoma compared to acral benign nevi [34]. Also in keratoacanthoma the moderate or strong Ki-67 staining was less likely to extend beyond the basal 1–3 cells of epithelium compared to squamous cell carcinoma [35]. In Tsunoda et al. case series of 13 EPC patients the Ki-67 index was 16.6% in patients with no lymph node metastases, 21.0% in patients with positive sentinel lymph nodes and, 32.3% in patients with clinical and/or revealed by imaging lymph node metastases at presentation [22]. In a smaller case series of EPC the Ki-67 index was described as clear proliferative activity but no accurate numbers were reported [36].

The aim of this study was to investigate the differences in clinicopathological factors between benign EP and malignant EPC with special interest in the Ki-67 index. The hypothesis was that the Ki-67 index is higher in EPC tumours compared to EP tumours as it has been reported being higher in malignant tumours compared to their benign counterparts. In addition, we studied the effect Ki-67 had on survival, tumour size and recurrences.

## Materials and methods

The Ethics Committee of Helsinki University Hospital approved the study plan and protocol. After individual approvals from each biobank, the tissue samples with corresponding clinical data were collected from three Finnish biobanks, the Helsinki Biobank (Helsinki), Finnish Clinical Biobank (Tampere), and Biobank Borealis (Oulu). We received 54 EP and 22 EPC samples from the biobanks. When combining the corresponding clinical data, 3 EP samples were found to be first misdiagnosed as eccrine poroma and later re-diagnosed as benign hidradenoma, benign pilomatrixoma and eccrine porocarcinoma. The one porocarcinoma was added to the EPC group and the other two excluded from the analysis. Within EPC group one sample was mislabelled as eccrine porocarcinoma and was actually carcinoma adenomatosum and excluded from the analysis. Finally, we had 51 EP and 22 EPC samples for analysis.

### Histopathological diagnosis of EP and EPC

In EP the growth pattern is solid and the tumour replace parts of the epidermis [37]. The tumour consists of small poroid cells with round to oval nuclei, which have basophilic cytoplasm and also larger cuticular cells that have more abundant eosinophilic cytoplasm and larger nuclei [38]. EPC resembles EP but shows signs of invasion and atypia of the tumour cells. The distinctive sign of EPC is the ductal formation within the tumour nests, which can be highlighted with carcinoembryonic antigen, epithelial membrane antigen and periodic acid-Schiff stain [2,16,21]. In this study all the samples received from the Biobanks were re-evaluated by a dermatopathologist.

### Immunohistochemistry

Formalin-fixed and paraffin embedded (FFPE) samples were constructed to tissue microarrays (TMA) by using 1.0 mm diameter core biopsy needle in the Helsinki Biobank and Biobank Borealis. Four µm sections were cut onto Labsolute microscope slides (Art. Nr. 7695015, The Geyer & Co, KG, Germany) followed by deparaffinization, rehydration and antigen retrieval (Tris-EDTA buffer, pH 9.0, 15 min at 98°C) for the tissues. Immunohistochemistry was performed as described by Valkonen et al. [39]. Shortly, tissues were incubated sequentially with the Ki-67 antibody (Dako-Agilent, clone MiB-1, 1:100, 30 min), anti-mouse peroxidase polymer (Histofine, Nichirei, 30 min) and 3,3'-diaminobenzidine. After this the slides were washed andcounterstained with hematoxylin and mounted with DPX mountant (Labvision Autostainer, Sigma-Aldrich).

### Ki-67 index analysis

Immunohistochemically stained objective slides were digitized into JPEG2000 file format as whole-slide images with Slide Strider instrument (Jilab Inc., Tampere, Finland) by scanning the slides under bright field with a x20 magnification lens with 0.16 µm per pixel. The images of slides were viewed with SlideVantage software that includes ImmunoRatio 2.5 program for Ki-67 proliferation rate image analysis Each patient had 1–4 TMA spots available for analysis. Within each TMA spot,

1–5 of the most representative tumour areas were manually selected and delineated by drawing regions of interest (ROIs) for the analysis (Fig. 1). In some tissue spots the tumour cell count was limited due to technical difficulties for example rolling of the slice. ImmunoRatio 2.5 calculated the number of positively stained nuclei per square millimetre and also the percentage of positively stained nuclear area [40]. The principle of this analysis is described in detail in previous studies and is based on the colour algorithm originally described by Ruifrok et al. [39–41].

We then calculated the mean values of Ki-67 count (cells/mm$^2$) and ratios (%) for each tumour. We used the means of each tumour for the statistical analysis.

## Statistical analysis

For all statistical analyses we used SPSS Software (IBM SPSS Statistics for Windows version 28.0. Armonk, NY: IBM Corp). All the continuous parameters were nonnormally distributed. For the comparison of EP and EPC groups we used Mann-Whitney U test for the continuous parameters and crosstabulation for categorical parameters (Fisher's exact test or Pearson $\chi^2$ test as appropriate). For the correlation assessment of nonnormally distributed continuous parameters, we used Spearman's correlation test. Overall survival analysis was carried out with Kaplan-Meier curve and log rank test and Cox regression analysis, by calculating survival from date of diagnosis to death and censoring patients who were alive on the date of data collection. Then all factors considered statistically significant (p value ≤0.05) were entered to Cox multivariate analysis and hazard ratios (HR) and 95% confidence intervals (CI) were indicated.

## Results

We included 51 EP and 22 EPC samples from 48 EP and 21 EPC patients. Within EP group two and EPC group one patient had two samples from the same tumour: the first was from the biopsy and the second from the complete removal of the tumour. In these patients we analysed only the first sample: the biopsy sample. Additionally, one EP patient had recurrence in short interval; the tumour recurred after two months of the first removal. In this patient we analysed the first tumour sample.

Within EPC group the distribution of females and males was equal, 48% and 52%, respectively. In the EP group there were more females (58% and 42%), but this difference was not statistically significant, p=0.442 (Table 1). The EPC patients were statistically significantly older; the median ages of EPC and EP groups were 74 years (range 19−92 years) and 68 years (range 8−95 years), p=0.019. Also, the EPC tumours were larger than EP tumours with size medians of

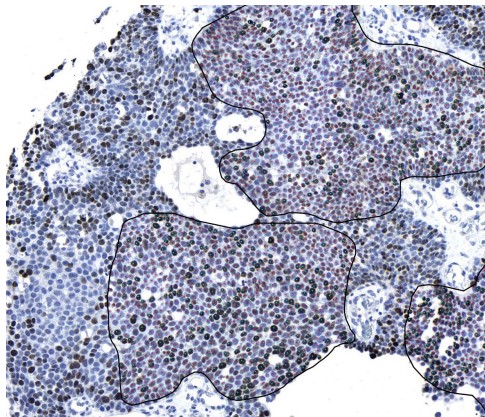

**Fig1. Example of selected tumour cell areas of eccrine porocarcinoma using the web based ImmunoRatio 2.5 program.** The Ki-67 index of this sample was 19%. Blue dotted tumour nuclei are stained with Ki-67 antibody and indicate cells in active phases of cell cycle. Red dotted nuclei do not express Ki-67.

**Table 1. Clinicopathological factors of eccrine poroma and porocarcinoma patients.**

| | Eccrine poroma | Missing data (n) | Eccrine porocarcinoma | Missing data (n) | Comparison of groups |
|---|---|---|---|---|---|
| **N** | **48** | | **21** | | |
| Sex | | | | | |
| female, n (%) | 28 (58%) | | 10 (48%) | | p = 0.442 |
| male, n (%) | 20 (42%) | | 11 (52%) | | |
| Age, years median (range) | 68 (8-95) | | 74 (19-92) | | p = 0.019 |
| Tumour size, mm median (range) | 10 (2-25) | 17 | 29 (5-150) | 8 | p = 0.003 |
| Tumour location[a], n (%) | | 5 | | | |
| Head and neck | 13 (30%) | | 7 (33%) | | p = 0.621 |
| Trunk | 9 (21%) | | 3 (14%) | | |
| Upper extremity | 8 (19%) | | 2 (10%) | | |
| Lower extremity | 13 (30%) | | 9 (43%) | | |
| Ki-67 number/mm$^2$, median (range) | 52 (0-422) | | 41 (0-1944) | | p = 0.265 |
| Ki-67%, median (range) | 0.6% (0-4.7%) | | 0.5% (0-19%) | | p = 0.667 |
| Local recurrence, n (%) | 2 | 43 | 2 (11%) | 3 | n/a |
| Metastases[b], n (%) | n/a | | 2 (11%) | 2 | n/a |
| Deaths, not specified, n (%) | 15 (34%) | 4 | 8 (40%) | 1 | p = 0.780 |

[a]Genitalia are included in the trunk, axilla in the upper extremity and inguinal area in the lower extremity.

[b]Metastases include skin, lymph node and distant metastases.

29 mm (range 5−150 mm) and 10 mm (range 2−25 mm), p = 0.003. In both groups the tumours located similarly across the body, most commonly in the lower extremities and head and neck area. There was no difference in the Ki-67 index between EP and EPC groups (Table 1). The Ki-67 ratio did not correlate to tumour size in either EP or EPC groups, p = 0.391 and 0.942, respectively (Spearman's rho correlation coefficients −0.163 and −0.022).

The information on local recurrences was available for EPC patients, of which 2 (11%) had a recurrence (Table 1). For EP patients the information on local recurrences was stated only in five patients and of those two had a recurrence. There was no statistically significant difference in the Ki-67 ratio in patients with or without recurrences in EPC group, p = 0.887 (Mann-Whitney U test). Two (11%) patients in the EPC group had metastasized disease and their Ki-67 ratios were 0% and 2%.

The median follow-up time was long for both EP and EPC groups; 12 (range 1.5–30.6 years) and 7 years (range 0.75–20.3 years), respectively. During this time 15 (34%) EP and 8 (40%) EPC patients died of unknown causes. The overall survival of EP patients was better than EPC patients according to Kaplan-Meier survival analysis (p = 0.055) (Fig. 2). Ki-67 index did not have statistically significant effect on survival in the Cox univariate regression analysis in the whole sample series (HR 0.883, 95% CI 0.657–1.186, p = 0.408) nor separately in the EP (HR 1.007, 95% CI 0.692–1.467, p = 0.97) or EPC groups (HR 0.746, 95% CI 0.390–1.43, p = 0.378). Tumour size had significant effect on survival both in the whole sample series (HR 1.034, 95% CI 1.012–1.057, p = 0.002) and in the EPC group (HR 1.031, 95% CI 1.001–1.062, p = 0.041) but not in the EP group (HR 0.981, 95% CI 0.853–1.102, p = 0.746). Age had a statistically significant effect on survival in both EP and EPC groups (HR 1.051, 95% CI 1.051–1.093, p = 0.011 for EP and HR 1.105, 95% CI 1.008–1.212, p = 0.033 for EPC).

When age and tumour type were put in the Cox multivariate analysis of the whole sample series, only age had a statistically significant effect on survival (HR 1.061, 95% CI 1.026–1.099, p < 0.001) but tumour type did not (EPC´s HR 1.933 relative to EP, 95% CI 0.761–4.909, p = 0.166). The size of the tumour was excluded from the analysis due to numerous missing size data points.

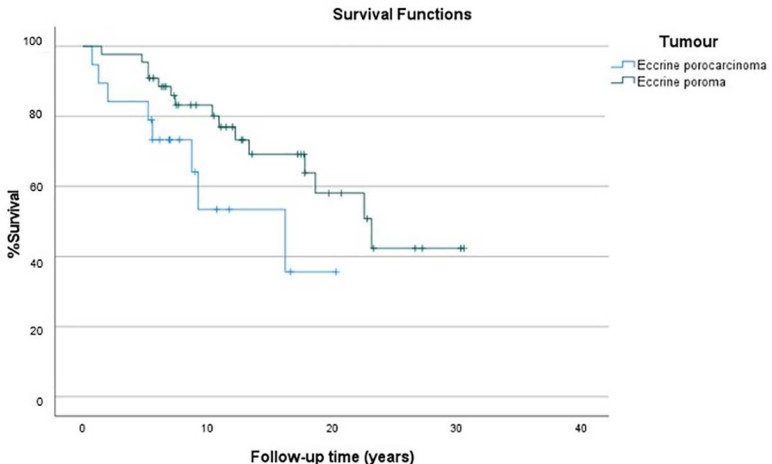

**Fig 2. The overall survival of eccrine poroma and porocarcinoma patients (Kaplan-Meier survival curve).**

## Discussion

Both EP and EPC are difficult to diagnose and more so to differentiate from one another. EP is found adjacent to EPC in histopathological samples in 2–27% [16,21,26,27], and it is debated whether all EPCs arise from benign EPs. Here we compared the clinicopathological characteristics and Ki-67 index between 48 EP and 21 EPC patients. In our study EP patients were younger than EPC patients which is also the trend in the literature [4–6,21,24–27]. EP tumours were smaller, but other than these two statistically significant differences the characteristics of both EP and EPC patients and tumours were quite similar. The clinical signs of malignant transformation of EP to EPC are ulceration and bleeding, tumour growth, itching and pain [42,43]. The malignant transformation takes place during decades [42,43] which might be one of the explanations for older age of EPC patients. In genetic studies both EP and EPC bear notable UV-induced mutational signatures, however, the total mutational load is higher in EPCs [44–46]. In a recent study EPCs carried mutations in *TP53*, *NCOR1* and *CDKN2A* which were not present in EPs and, EPCs had mutations in pathways associated with cell adhesion and the extracellular matrix [46].

The recurrence rates of EP were difficult to find from the literature, but 4–22% of EPC patients had local skin recurrences [10,14,21,27] which is in line with our rate (11%). During follow-up 11% of EPC patients the disease metastasized which corresponds to the rates in the recent literature [27,28]. In our series the survival of EP patients was better than EPC patients but when analysing whole series in the Cox multivariate analysis only age had a statistically significant effect on survival, but tumour histology did not. EPC patients were statistically significantly older than EP patients which could explain the better survival of EP patients. This is also in line with previous epidemiological study from Finland in which the survival of EPC patients was no worse than the general population [47].

Surprisingly in our series the Ki-67 index did not have statistically significant difference when comparing EP and EPC groups which differs from the previous literature. Ansai et al. compared 25 benign eccrine tumours to 25 of their malignant counterparts and reported that the Ki-67 index was higher in the malignant group [48]. In another study by Pozo et al. the Ki-67 index was statistically higher in EPC (20%) than in EP (5%) but interestingly, EP had one of the highest Ki-67 index within the benign group of skin tumours with ductal differentiation [49]. Also, in Battistella et al. study the Ki-67 index of poroma was low; <2% in 36% and 2–5% in 64% of the samples [38]. However, in 36% of the samples there was very local high Ki-67 index values of 10–15% [38]. The Ki-67 index showed no differences within the malignant eccrine skin tumour group when comparing combined stages I and II to more aggressive stage III [48]. In our study the Ki-67 index did not

have a statistically significant effect on survival in the whole population nor in EP or EPC groups. Although Tsunoda et al. in their case series of 13 EPC patients did not report survival analysis, there was a trend showing that the Ki-67 index was higher in metastasized tumours [22]. Altogether, the Ki-67 index in our study was lower compared to previous literature [22,48,49] which could be explained by the differences in pathology laboratory protocols.

To our knowledge this is a unique study comparing the clinicopathological features of EP and EPC patients. The strengths of this study include that considering the rarity of these tumours we had good sample sizes of 48 EP and 21 EPC patients whose follow-up times were long (medians of 12 years and 7 years, respectively). Although our Ki-67 index values were lower compared to the literature we could make reliable comparison between the two groups since all the immunohistochemistry was done from the samples collected from the Biobanks by the same protocol by our research team.

To conclude, it seems that in EP patients the survival is better than in EPC patients, but tumour histology was not an independent statistically significant factor. In our study the Ki-67 index showed no difference between EP and EPC groups and had no statistically significant impact on tumour size, recurrence, or survival. It might be that Ki-67 is not a deciding factor in EPC tumour progression or aggressiveness as in other cancer types such as breast cancer and neuroendocrine carcinomas of the digestive system where it is part of the routine diagnostic work-up [33]. In the light of our study Ki-67 index is not a useful prognostic tool in clinical practice. The role of Ki-67 index in EPC remains to be elucidated and needs further studies.

## Author contributions

**Conceptualization:** Anna-Stiina Meriläinen, Harri Sihto, Virve Koljonen.

**Formal analysis:** Anna-Stiina Meriläinen.

**Investigation:** Anna-Stiina Meriläinen.

**Methodology:** Anna-Stiina Meriläinen, Harri Sihto, Virve Koljonen.

**Software:** Jorma Isola.

**Supervision:** Harri Sihto, Virve Koljonen.

**Writing – original draft:** Anna-Stiina Meriläinen.

**Writing – review & editing:** Harri Sihto, Jorma Isola, Virve Koljonen.

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
