## [Decision Letter · Decision Letter 0]

10 Mar 2025

PONE-D-24-44914Clinicopathological comparison of eccrine poroma and porocarcinoma: Ki-67 index is not a decisive factorPLOS ONE

Dear Dr. Anna-Stiina Meriläinen,

Thank you for submitting your manuscript to PLOS ONE. After careful consideration, we feel that it has merit but does not fully meet PLOS ONE’s publication criteria as it currently stands. Therefore, we invite you to submit a revised version of the manuscript that addresses the points raised during the review process.

Please submit your revised manuscript within  Apr 24 2025 11:59PM. If you will need more time than this to complete your revisions, please reply to this message or contact the journal office at plosone@plos.org . .

For Lab, Study and Registered Report Protocols: These article types are not expected to include results but may include pilot data. If applicable, we recommend that you deposit your laboratory protocols in protocols.io to enhance the reproducibility of your results. Protocols.io assigns your protocol its own identifier (DOI) so that it can be cited independently in the future. For instructions see: https://journals.plos.org/plosone/s/submission-guidelines#loc-laboratory-protocols . Additionally, PLOS ONE offers an option for publishing peer-reviewed Lab Protocol articles, which describe protocols hosted on protocols.io. Read more information on sharing protocols at https://plos.org/protocols?utm_medium=editorial-email&utm_source=authorletters&utm_campaign=protocols .

We look forward to receiving your revised manuscript.

Kind regards,

Dr H. Boukerche, PhD

Academic Editor

PLOS ONE

Journal Requirements:

a) If there are ethical or legal restrictions on sharing a de-identified data set, please explain them in detail (e.g., data contain potentially identifying or sensitive patient information, data are owned by a third-party organization, etc.) and who has imposed them (e.g., a Research Ethics Committee or Institutional Review Board, etc.). Please also provide contact information for a data access committee, ethics committee, or other institutional body to which data requests may be sent

Reviewers' comments:

Reviewer's Responses to Questions

**Comments to the Author**

1. Is the manuscript technically sound, and do the data support the conclusions?

Reviewer #1: Yes

Reviewer #2: No

2. Has the statistical analysis been performed appropriately and rigorously? 

Reviewer #1: Yes

Reviewer #2: Yes

3. Have the authors made all data underlying the findings in their manuscript fully available?

Reviewer #1: Yes

Reviewer #2: No

4. Is the manuscript presented in an intelligible fashion and written in standard English?

Reviewer #1: Yes

Reviewer #2: Yes

5. Review Comments to the Author

Reviewer #1: The manuscript presents a study comparing clinicopathological factors between eccrine poroma (EP) and porocarcinoma (EPC), with a special focus on the Ki-67 index. The research is well-structured, addresses an important and rather underexplored topic, relevant statistical analysis has been applied.

The abstract provides a good overview but should also emphasize the clinical implications demonstrated by this study. Consider adding future research plans, if any are foreseen.

The introduction is elaborate, scientifically well-supported by the relevant references. I would recommend adding more explanation the hypothesis here why Ki-67 was expected to differ between EP and EPC as it is somewhat elaborated only in the discussion section.

Methods are reasonably described. However, the Ki-67 index analysis could be described even in a bit more detailed manner - for the audience not to be looking into the references 35 and 36. Taking into consideration the rarity of the disease, sample collection and potential biases in the process, sample size calculation / power analysis could also be explained to strengthen the study’s validity.

Results and statistical findings are well-described. Consider discussing the high age in EPC patients as a possible confounding factor and if/how it may have influenced survival presented in Kaplan-Meier curves.

The discussion is interesting to read and reasonably supported by citing other studies. Consider emphasizing the clinical relevance of your findings, e.g., should Ki-67 be used in routine practice to prognose EPC?

Best of luck!

Reviewer #2: The Ki-67 positivity rate in eccrine porocarcinoma (EPC) has been reported. The authors stated that the Ki-67 positivity rate was 0.6% in eccrine poroma (EP) and 0.5% in EPC, indicating no significant difference between the two tumor groups. Although I had expected a slightly higher rate in EPC, the actual rate was not as high.

Whole-exome sequencing identifies distinct genomic aberrations in eccrine porocarcinomas and poromas

To ensure the scientific rigor of the paper, please include the pathological diagnostic criteria for EP and EPC in the Methods section. How about including pathological images of all specimens stained with HE and Ki-67 as supplemental data?

Reviewer #3 : As an academic editor,I would suggest it would be important to discuss these studies in the context of several studies such as the one below that shows distinct genomic aberrations in eccrine porocarcinomas and poromas including *TP53* , *NCOR1* , and *CDKN2A.*

                                                                                                                                                                                                                                                                                 Whole-exome sequencing identifies distinct genomic aberrations in eccrine porocarcinomas and poromas.  Puttonen M, Almusa H, Böhling T, Koljonen V, Sihto H. **Orphanet J Rare Dis.** 2025 Feb 13;20(1):70.

6. PLOS authors have the option to publish the peer review history of their article (what does this mean? ). If published, this will include your full peer review and any attached files.

**Do you want your identity to be public for this peer review?** For information about this choice, including consent withdrawal, please see our Privacy Policy .

Reviewer #1: No

Reviewer #2: **Yes: ** TERUKI YANAGI

---

## [Author Response · Author response to Decision Letter 1]

24 Apr 2025

Authors’ responses to the Reviewers’ comments:

Reviewer 1:

The manuscript presents a study comparing clinicopathological factors between eccrine poroma (EP) and porocarcinoma (EPC), with a special focus on the Ki-67 index. The research is well-structured, addresses an important and rather underexplored topic, relevant statistical analysis has been applied.

1. The abstract provides a good overview but should also emphasize the clinical implications demonstrated by this study. Consider adding future research plans, if any are foreseen.

RE: The abstract has been modified according to suggestion.

2. The introduction is elaborate, scientifically well-supported by the relevant references. I would recommend adding more explanation the hypothesis here why Ki-67 was expected to differ between EP and EPC as it is somewhat elaborated only in the discussion section.

RE: We have now added explanation in the Introduction as advised.

3. Methods are reasonably described. However, the Ki-67 index analysis could be described even in a bit more detailed manner - for the audience not to be looking into the references 35 and 36. Taking into consideration the rarity of the disease, sample collection and potential biases in the process, sample size calculation / power analysis could also be explained to strengthen the study’s validity.

RE: We have now added the immunohistochemical staining in more detail in the Methods-section. As the disease under investigation is extremely rare, our study included all eligible cases available during the study period. Therefore, no prospective sample size calculation or formal power analysis was performed. We acknowledge this as limitation. Nonetheless, we believe the collected data provide meaningful insights into this rare condition and serve as a valuable contribution to the limited literature available.

4. Results and statistical findings are well-described. Consider discussing the high age in EPC patients as a possible confounding factor and if/how it may have influenced survival presented in Kaplan-Meier curves.

RE: This is now discussed in the Discussion more elaborately.

5. The discussion is interesting to read and reasonably supported by citing other studies. Consider emphasizing the clinical relevance of your findings, e.g., should Ki-67 be used in routine practice to prognose EPC?

RE: We have added a suggestion of the use of Ki-67 index in clinical practice in the Discussion.

Reviewer 2

1. Whole-exome sequencing identifies distinct genomic aberrations in eccrine porocarcinomas and poromas

To ensure the scientific rigor of the paper, please include the pathological diagnostic criteria for EP and EPC in the Methods section. How about including pathological images of all specimens stained with HE and Ki-67 as supplemental data?

RE: We have added a section describing the histopathological diagnosis of EP and EPC in the Methods-section. We are not allowed to include pathological images of all specimens as supplemental data due to the strict legislation of Finnish Biobanks.

Reviewer 3

1. As an academic editor, I would suggest it would be important to discuss these studies in the context of several studies such as the one below that shows distinct genomic aberrations in eccrine porocarcinomas and poromas including TP53, NCOR1, and CDKN2A.

Whole-exome sequencing identifies distinct genomic aberrations in eccrine porocarcinomas and poromas. Puttonen M, Almusa H, Böhling T, Koljonen V, Sihto H. Orphanet J Rare Dis. 2025 Feb 13;20(1):70.

RE: We have included the suggestions in the Discussion

---

## [Editor Report · Decision Letter 1]

5 May 2025

Clinicopathological comparison of eccrine poroma and porocarcinoma: Ki-67 index is not a decisive factor

PONE-D-24-44914R1

Dear Dr. Anna-Stiina Meriläinen

We’re pleased to inform you that your manuscript has been judged scientifically suitable for publication and will be formally accepted for publication once it meets all outstanding technical requirements.

Kind regards,

Dr H Boukerche, PhD

Academic Editor

PLOS ONE
---

## [Editor Report · Acceptance letter]

PONE-D-24-44914R1

PLOS ONE

Dear Dr. Meriläinen,

I'm pleased to inform you that your manuscript has been deemed suitable for publication in PLOS ONE. Congratulations! Your manuscript is now being handed over to our production team.

Kind regards,

on behalf of

Dr Habib Boukerche

Academic Editor

PLOS ONE